behaviour/biophysics

cooperation, working memory, evolutionary game theory

**Authors for correspondence:**
Boyu Zhang
e-mail: zhangby@bnu.edu.cn
Wen-Xu Wang
e-mail: wenxuwang@bnu.edu.cn

# Limited memory optimizes cooperation in social dilemma experiments

Shuangmei Ma[1,4], Boyu Zhang[2], Shinan Cao[5], Jun S. Liu[4] and Wen-Xu Wang[1,3]

[1]School of Systems Science, [2]Laboratory of Mathematics and Complex Systems, Ministry of Education, School of Mathematical Sciences, and [3]State Key Laboratory of Cognitive Neuroscience and Learning and IDG/McGovern Institute for Brain Research, Beijing Normal University, Beijing 100875, People's Republic of China
[4]Department of Statistics, Harvard University, Cambridge, MA 02138, USA
[5]School of Finance, University of International Business and Economics, Beijing 100029, People's Republic of China

(iD) BZ, 0000-0003-0972-3240; W-XW, 0000-0002-4170-8676

Cooperation is one of the key collective behaviours of human society. Despite discoveries of several social mechanisms underpinning cooperation, relatively little is known about how our neural functions affect cooperative behaviours. Here, we study the effect of a main neural function, working-memory capacity, on cooperation in repeated Prisoner's Dilemma experiments. Our experimental paradigm overcomes the obstacles in measuring and changing subjects' working-memory capacity. We find that the optimal cooperation level occurs when subjects remember two previous rounds of information, and cooperation increases abruptly from no memory capacity to minimal memory capacity. The results can be explained by memory-based conditional cooperation of subjects. We propose evolutionary models based on replicator dynamics and Markov processes, respectively, which are in good agreement with experimental results of different memory capacities. Our experimental findings differ from previous hypotheses and predictions of existent models and theories, and suggest a neural basis and evolutionary roots of cooperation beyond cultural influences.

## 1. Introduction

Cooperation is essential to human society and is key to the success of human species in biological evolution. However, human cooperation in many cases contradicts model predictions based on the self-interest assumption, as exemplified by the Prisoner's Dilemma (PD) game and the public goods game [1–5]. To resolve the conflict, a variety of mechanisms accounting for the emergence

of cooperation have been proposed, such as reciprocity [6–9], reputation [10–13], inequality aversion [14,15], punishment [16–19], etc. Most of the proposed mechanisms are related to social norms and are the consequence of cultural evolutions in human society [13,20,21]. A large proportion of the mechanisms has been experimentally validated, but a minority remains to be confirmed. Despite the progress in understanding cooperation in terms of social influence, relatively little is known about how our biological basis affects human cooperation among genetically unrelated individuals [22]. In fact, the uniqueness of humans on earth stems from the joint evolution of biology and culture [23]. To comprehensively understand human cooperation, it is imperative to take neural functions of the human brain into account rather than only focusing on social effects [24–26].

Accompanied with the development of behavioural economics, it is realized that people are bounded rational, differing from the classical hypothesis of rational agents in economy [27]. The bounded rationality may stem from many aspects, such as cognitive biases, social norms, limited information and finite computation ability of the brain [28,29]. As a consequence, human behaviours deviate from the assumption of pursuing maximum profits [30]. Concerning the neural basis of making decisions, working memory plays a key role in bounded rational behaviours of humans. Social interactions, reputation recognition, calculation of profits and many other capabilities rely crucially on remembering critical but finite information [31]. Memory capacity also determines social organization of human and animals [32], e.g. the size of communities is limited by the Dunbar number [33]. Even in modern society with the Internet and efficient communication equipment the limit of the Dunbar number still holds [33]. The Dunbar number is mainly attributed to the limited memory capacity for remembering and keeping relation with community members. Working memory is a significant issue in cognitive neuroscience and psychology [34]. Despite the existent dispute about the mode of allocating neural resources in working memory [35,36], the fact that working-memory capacity is limited to about 4–7 items is generally accepted [37–39].

At present, how limited memory capacity affects human cooperation is still an outstanding question. Several models have been introduced to predict the effect of memory capacity via evolutionary game theories and simulations [40–46]. A few debates arise about whether a larger memory capacity leads to a higher cooperation level in a population. On the one hand, evolutionary game studies based on direct reciprocity suggest that a longer memory can lead to a higher cooperation level in the repeated social dilemma game [40,44,45]. On the other hand, numerical simulations based on network effect and indirect reciprocity showed that intermediate memory length could facilitate the emergence of cooperation [42,46]. Experimental tests are necessary to resolve these disputes, but encounter two significant difficulties. The first one is that the innate variation of memory capacity of subjects is often associated with other mental deficiencies, making it difficult to assess the effect of memory capacity on cooperation without severe confounding. Second, it is difficult to know how and how much temporal information in subjects' memories is used for making a decision. Because of these obstacles, few experiments about memory effect on cooperation have been conducted and the debate of memory effect on cooperation continues [41,47].

We here introduce an experimental paradigm to explore the effect of working-memory capacity on cooperation. The paradigm allows us to guarantee identical working-memory capacity of subjects with normal intelligence and to adjust their capacities simultaneously and arbitrarily. The experiments revealed that strategies of subjects for making decisions based on limited working-memory capacity differ from those in existent models in the literature, and memory-based conditional cooperation (MC) plays a significant role. Our results provide credible evidence that bridges the biological basis associated with bounded rationality and prosocial behaviours in the population.

## 2. Results

### 2.1. Experimental setting

In principle, we control the memory capacity of subjects in a group by anonymous interactions. The experiment includes 150 rounds of repeated PD game, and in each round, a subject is informed that she/he randomly interacts with another subject in a group anonymously. There are two possible choices (or actions) for each player: cooperate (C) or defect (D), and the payoff matrix is

|   | C | D |
|---|---|---|
| C | 2 | 0 |
| D | 3 | 1 |

The key to the control lies in the feedback information provided to subjects for making decisions. In particular, for two arbitrary subjects, say $i$ who plays with subject $j$ in the current round, only the actions of $j$ previously used to play with $i$ and the actions of $i$ previously used to play with $j$ are shown to $i$. If the memory capacity (or lengths) of subject $i$ is $L$, the feedback information is the $L$-round historical strategies, which corresponds to $i$'s own actions and $j$'s actions in the specified $L$ latest rounds. Because interactions in a group are randomized in each round, it is impossible to track anonyms. Thus, for subject $i$ the only information that can be used to play with $j$ is $i$'s and $j$'s specified $L$-round historical actions (Methods and electronic supplementary material, appendix, figure S1). In other words, subjects' memories are reloaded with an adjustable amount of historical information. This design guarantees that all subjects' memory capacities can be adjusted, regardless of their actual memory capacities. More detailed experimental settings can be found in Methods and electronic supplementary material, appendix, §1.

A total of 368 university students participated in the experiment. They were divided into five treatment sessions. In each session, a group consists of eight subjects and there are in total 46 groups in five sessions. It is worth noting that the experiment is essentially a two-player game. The purpose of implementing group experiments is to better control the memory capacity. The memory capacities for subjects in the five treatment sessions named T0–T4 are $L = 0$, $L = 1$, $L = 2$, $L = 3$ and $L = 4$, respectively. In particular, when there is no historical information provided to subjects, it is not possible for subjects to track anonyms under random interaction, accounting for the absence of memory (the memory length is 0).

## 2.2. Experimental results

To avoid the end-game effects, we use the experiment data from round 1 to round 140 for analysis. Figure 1$a$ shows the time evolution of collective cooperation levels. We see that the cooperation rate decreased initially from about 0.5 for all sessions. After a few rounds, in the absence of memory, the cooperation rate in T0 deviated from those of other sessions and decreased continuously to about 0.1 eventually. By contrast, cooperation rates associated with T1–T4 gradually increased and exceeded 0.5 in all the sessions finally. This difference demonstrates that short-term memory is imperative to promote and sustain cooperation. The low cooperation rate in T0 is consistent with the classical prediction based on game theory, in the sense that the absence of memory hinders reciprocity between subjects. The initial decrease of cooperation rates in all sessions may be ascribed to the lack of sufficient information to form reciprocity temporally among subjects.

Figure 1$b$ shows stable cooperation rates of the treatments during the second half rounds. A prominent result is that the cooperation rate peaked at T2, and the peak value of about 0.6 is significantly higher than those of T0 and T4 (Wilcoxon rank-sum test, $p$-value $< 0.05$ for each pair of comparisons; electronic supplementary material, appendix, table S1). This indicates that intermediate memory length optimizes cooperation between two subjects. This phenomenon has never been experimentally discovered. It is somewhat counterintuitive that a higher memory capacity with more information of partners could render the cooperation rate lower. Because the classical evolutionary model is unable to explain the non-monotonic cooperation rate as a function of memory length [44,45], we speculate that some psychological effect may play a role in making decisions. To understand possible mechanisms underlying the memory effect, we study individual behaviours and seek a simple evolutionary model.

We analyse individuals' behaviours in T1–T4 in the framework of reactive strategies [4,9,40,45,48], according to which the decision of an individual in the current round is determined by outcomes in previous rounds with his/her current partner. Reactive strategies in T1–T4 are summarized in electronic supplementary material, appendix, table S2. In all the treatment sessions, we find that each individual's behaviours when encountering an opponent depend strongly on the numbers of historical C and D decisions made by opponent revealed to the individual, where the probabilities that subjects choose C increase as their partner's frequency of C in memories increases (figure 1$c$; electronic supplementary material, appendix, figure S2). Furthermore, despite the complete cooperation history of two subjects, they could still switch to defection in the future with a small probability, where the switch rate decreases log-linearly as memory length increases, as shown in figure 1$d$. This observation reflects the stability of mutual cooperation: a longer memory of the opponent's history of cooperation without defection reinforces the subject's confidence of future cooperation. In addition, the switch rate from D to C of subjects with complete defection history in T1 is significantly higher than T2–T4, which implies that re-establishing cooperation is more difficult for a longer memory length ($t$-test; see electronic supplementary material, appendix, table S3 and table S4).

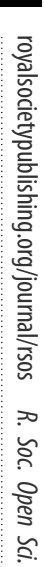

**Figure 1.** Cooperation rates and individual behaviours in the 5 experimental sessions. (*a*) Cooperation rate as a function of round in 5 experimental sessions. The shadow of lines represents mean ± s.e.m. (*b*) Average cooperation rates over the last 70 rounds in the 5 sessions. (*c*) The relationship between cooperation rates and the partner's frequency of C in the memory. When there is feedback information, the probabilities that subjects choosing C are increasing with the partner's frequency of C in their memories. (*d*) The relationship between switch rates and memory length. The switch rate is defined as the probability of choosing D when there is no D in the memories. Since the switch rate is a log-linear function of the memory length $L$, we formulate the switch rate by $\sigma^L$ in the theoretical model. Data points in (*a*) are taken as the smoothing of 5 rounds and in (*b*) *, ** and *** of the statistical test represent *p*-value less than 0.05, 0.01 and 0.001, respectively. Error bar shows the s.e.m.

To better understand how subjects make decisions, we further classify the subjects into four types based on their behaviours, i.e. always cooperation (AC), always defection (AD), MC and Other (more details in Methods), where an AC subject always chooses C, an AD subject always chooses D and a MC subject chooses C if his/her partner's cooperation rate is more than 0.5. We find 70–80% of subjects using MC (electronic supplementary material, appendix, table S5). To better understand the motivation and characteristics of MC subjects, we implement a logistic regression. The results show that the behaviour of MC subjects is mainly affected by their partners' behaviours, analogous to the definition of conditional cooperation. Meanwhile, the MC subjects exhibit selfishness in the sense that they occasionally violate mutual cooperation with their partners to gain more profits. The minor violation of conditional cooperation is measured by the small switching rate in the mutual cooperation state. Other factors, such as indirect reciprocity, have no significant effect on the behaviour of the MC player according to the regression (electronic supplementary material, appendix, table S6). Taken together, we build an evolutionary model with regard to the conditional cooperation and selfishness of the MC players and the competition among AC, AD and MC categories of players to reproduce experiment results and test our hypothesis.

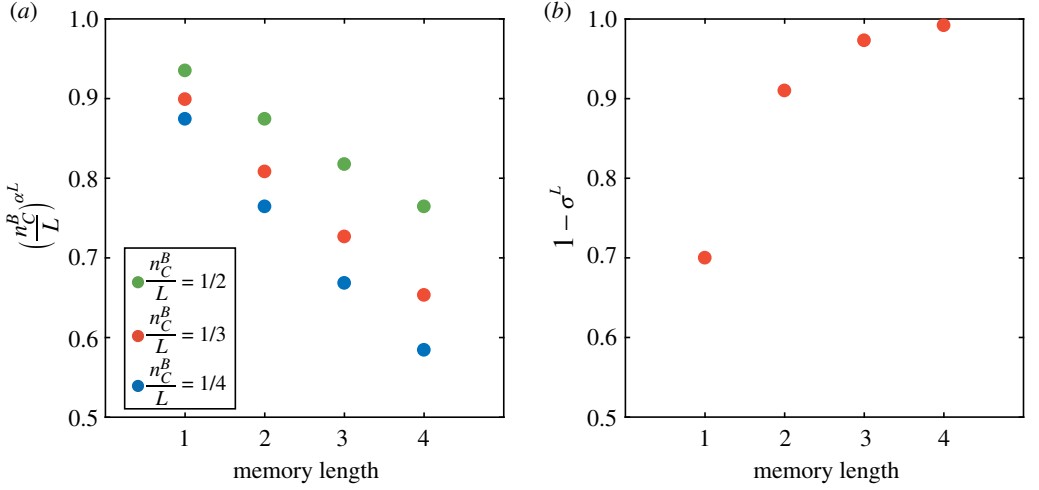

**Figure 2.** MC. The cooperation rate of the MC player A interacting with MC player B is formulated as $P_C^A = (n_C^B/L)^{\alpha L}(1 - \sigma^L)$, where $\alpha = 0.0969$ and $\sigma = 0.3001$. (a) $(n_C^B/L)^{\alpha L}$ reflects the effect of conditional cooperation. A longer memory has a negative effect on cooperation because a defective behaviour is more difficult to forget. (b) $1 - \sigma^L$ reflects the stability of mutual cooperation, where the stability is enhanced as memory length increases.

## 2.3. Theoretical model

We assume that a subject in the repeated PD game can use only one of the three possible strategies: AC, AD and MC, and the fractions of players adopting these strategies are denoted as $\rho_{AC}$, $\rho_{AD}$ and $\rho_{MC}$, respectively [4,49]. In each round of the repeated PD game, AC and AD players choose C and D, respectively, regardless of the historical information. By contrast, decisions of MC players depend on the cooperation rate of the partner and the switch rate to defection. Specifically, a MC player A will choose C when encountering an AC player, and D when encountering an AD player, and choose C with probability $P_C^A = (n_C^B/L)^{\alpha L}(1 - \sigma^L)$ when encountering another MC player, B, where $n_C^B$ is the number of historical C that the opponent B used with A in the memory, and $\sigma^L$ is the switch rate to defection in the case of mutual cooperation (figure 1d), and $\alpha$, $\sigma$ are two free parameters. The first part, $(n_C^B/L)^{\alpha L}$, reflects the effect of conditional cooperation, where $\alpha \in (0, 1)$. It captures the fact that for the same proportion of positive memories $n_C^B/L$, a longer memory length has a stronger negative effect on cooperation. In all the treatments, the improvement of cooperation rate due to the increase of $n_C^B$ is significantly less than the reduction of cooperation rate due to the decrease of $n_C^B$ (electronic supplementary material, appendix, figure S3). This implies that the negative and positive memories have asymmetrical influence and people are more sensitive to previous defection. For the same value of $n_C^B/L$, as the memory length increases, the amount of positive and negative memories increases simultaneously. But the stronger effect of the negative memories in subjects reduces their cooperation intensions. For simplicity, we use a power function to capture the nonlinear memory effect and facilitate our theoretical analysis. As shown in figure 2a, indeed, the power function $(n_C^B/L)^{\alpha L}$ is able to characterize the asymmetry between positive and negative memories, and the decrease of cooperation tendency as memory length augments. The second part, $1 - \sigma^L$, reflects the stability of mutual cooperation, where the stability is enhanced as memory length increases (figure 2b). $\sigma^L$ is the switching rate of a subject in a mutual cooperation state and reflects the selfishness of subjects in terms of gaining extra profits by occasional betrayal (figure 1d). However, the switching behaviour incurs a risk of ruining the cooperation alliance with partners. As a result, subjects will weigh their choices between acquiring risky payoffs or keeping a healthy cooperation pair. In general, in a more stable mutual cooperation pair, maintaining the social tie outweighs risky profits. In addition, the switching rate associated with the instability of mutual cooperation might be due to the fact that a shorter memory allows less learning about partner types. By contrast, a longer memory allows subjects to better identify their partner types and facilitate conditional cooperation. As a consequence, a longer memory accounts for more stable mutual cooperation with a lower switching rate. Since the two effects are independent, we combine them multiplicatively. Note that the first part $(n_C^B/L)^{\alpha L}$ and the second part $1 - \sigma^L$ are monotonically decreasing and increasing functions of $L$, respectively. Their combination yields a non-monotonic function of $L$. Insofar as the cooperation probabilities $P_C^A$

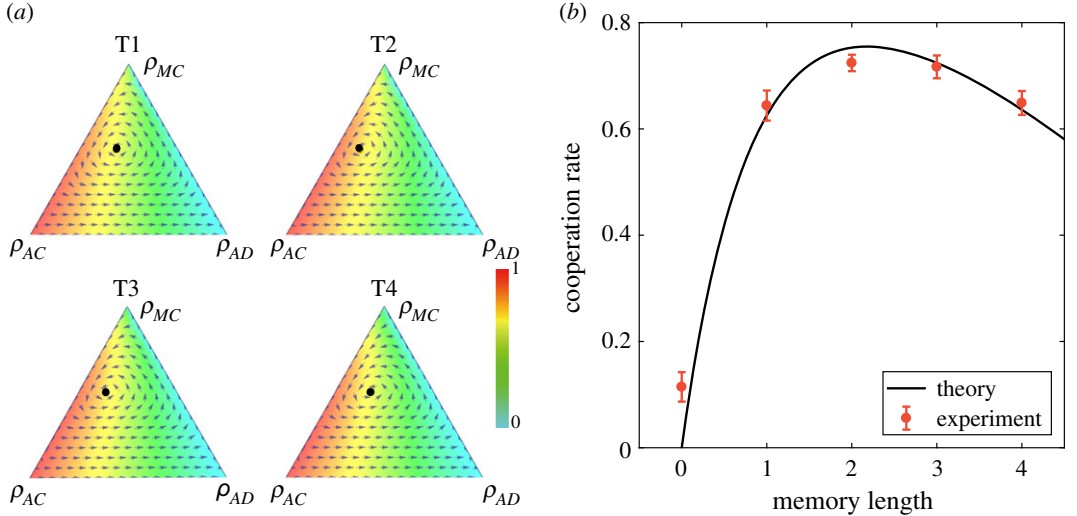

**Figure 3.** Comparison between experimental data and our equilibrium theory. (*a*) Theoretical phase portraits of the replicator equations in T1–T4 with $\alpha = 0.0969$, $\sigma = 0.3001$. The black points denote the stable interior equilibrium. (*b*) Comparison between experimental results and equilibrium points obtained by the evolutionary model. The black curve is obtained from the formula of interior equilibrium of the replicator equations. The red points are the experimental data obtained by the average of the last 70 rounds. Error bar shows the s.e.m.

and $P_C^B$ between two MC players are ascertained, a payoff matrix of AC, AD and MC can be established (electronic supplementary material, appendix, §2). The payoff matrix allows us to formulate the evolutionary dynamics of cooperation affected by limited memory capacity.

We note that subjects in the experiments may not know the types of their partners, because a real subject may use a mixture of AC, AD and MC. In fact, our model only describes the behaviours of players that use only one of the three pure strategies. Real-life situations are likely more complicated.

We apply replicator equations to study the frequencies of subjects using AC, AD and MC strategies at the stable state. Replicator equations assume that subjects are boundedly rational and tend to use the strategies with payoffs higher than the group average [50]. To investigate how memory length affects the coexistence of C and D, we calculate the stable interior equilibrium of the replicator equations, where at this equilibrium all the three strategies have the same expected payoff (electronic supplementary material, appendix, §4 for details of the replicator equations). We theoretically identified the existence of a unique interior equilibrium point: $(\rho_{AC}, \rho_{AD}, \rho_{MC}) = ((2/1 - \sigma^L)^{1/\alpha L-1}, 1/2 - (1/1 - \sigma^L)^{1/\alpha L-1}, 1/2)$. The average cooperation rate at this equilibrium is $(2^{\alpha L}/1 - \sigma^L)^{1/\alpha L-1}$.

The interior equilibrium has two independent free parameters $\alpha$ and $\sigma$. It is difficult to directly compare the theoretical equilibrium with individual strategies in the experiments because subjects may use mixed strategies. Alternatively, we estimate these parameters by comparing the analytical results of collective cooperation rate at the predicted equilibrium with the experimental cooperation rate. By applying the least square method, we obtained $\alpha = 0.0969$, $\sigma = 0.3001$.

Figure 3*a* shows the interior equilibrium point in the phase diagram and the cooperation rate associated with all the possible combination of three strategies. We see that the variation of the memory length induces the motion of the equilibrium point and changes in the distribution of collective cooperation rate in the phase diagram. The joint changes in the phase diagram affected by memory length account for the highest cooperation in T2. We have tested the stability of the equilibrium based on stability analysis (electronic supplementary material, appendix, §4). Specifically, the equilibrium is stable if and only if $1 - \alpha L > 0$. For all the sessions $L \leq 4$, the equilibrium is stable. The stability is also immediately reflected in the manifold toward the equilibrium point in the phase diagram in figure 3*a* as we can see that the 'flows' are towards the equilibriums. Figure 3*b* shows the comparison between the analytical results and experimental observation of cooperation rates. The analytical results are in good agreement with experiments with an optimal cooperation at treatment T2. Furthermore, the analytical results explicitly predict the decrease of cooperation rate for memory longer than 2.

Beyond stationary states modelled by the replicator dynamics, we also study transient behaviours of subjects in the experiments by means of a Markov model to reveal transient dynamics that cannot be

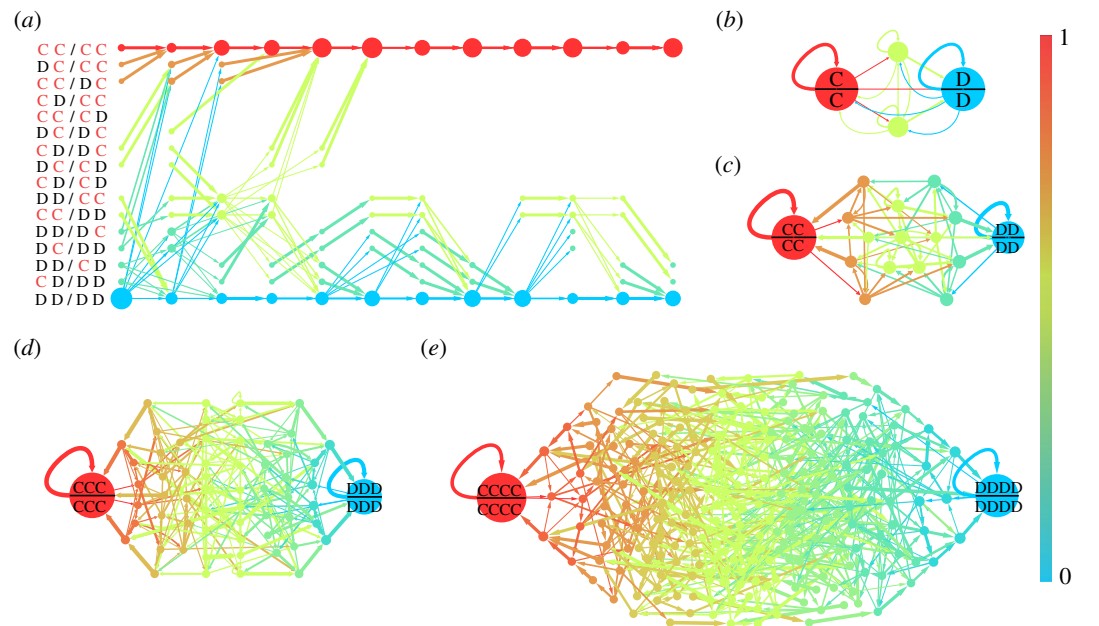

**Figure 4.** Transition graphs of historical states. (*a*) The multiple-layer graph of time-variant transition matrices of memory-2 experiments. Each layer aggregates 10 rounds, where the first layer represents rounds 15–24 (in T2, subjects obtain full memory after 14 rounds). The size of each node in a layer represents the average fraction of historical states in the 10 rounds, and the arrowed links represents the average transition probabilities. (*b*–*e*) Static transition graphs of memory-1 (*b*), -2 (*c*), -3 (*d*) and -4 (*e*) by collapsing multiple-layer graphs into a single layer. Here node sizes represent the fraction of historical states, colours represent the proportion of strategy C in a historical state, and the width of arrowed links represents transition probabilities. Self-loops represent the self-transition of states. An explicit polarization phenomenon is manifest in all of the transition graphs.

captured by the replicator dynamics. We demonstrate below that the evolutionary dynamics in the experiments can be effectively captured by a Markov process, and the Markov model endowed with transfer matrices offers a good prediction of temporal behaviours. First, we define a memory-*L* state as the joint historical actions of two encountered players with memory capacity *L*. Let us take two players, Alice and Bob, to illustrate the memory-2 states. The historical information got by Alice at time *t* can be written as a 4-dim vector $S_t = (A_{t-2}A_{t-1}/B_{t-2}B_{t-1}) = $ (Alice's move at $t-2$ and $t-1$ /Bob's move at $t-2$ and $t-1$). Thus, there are totally $2^4 = 16$ possible historical states: CC/CC, DC/ CC, etc. as shown in figure 4*a*. Analogously, there are $2^6 = 64$ states for memory-3 and $2^8 = 256$ states for memory-4. Transitions between these states from one round to another can be counted in the experimental data.

In a memory-*L* game, the state transition between two consecutive rounds can be depicted by a transition graph with directed and weighted links to represent the transition matrix, which is time invariant. Figure 4*a* exemplifies the transition graph estimated from the experimental data of a memory-2 group. The stationary distribution of a time invariant Markov chain typically exhibits a polarized transience phenomenon, i.e. nearly all initial states converge to all C and all D states eventually, if the transition matrix processes 'attractor states'. The polarization phenomenon is prominent in all experiments with different memory lengths, as shown in figure 4*b*–*e*. They show two big attractors, all C and all D, and all of the other temporal states transition to either one of them finally. The likelihood of escaping from the two attractors is very small, rendering states polarized and cooperation rates stable.

The transition matrix *M* for characterizing transition between states can be obtained based exclusively on MC. Specifically, when two subjects encounter, the probability of cooperation and defection for each of them is determined by the historical actions of the other side according to MC in figure 1*c*. Consequently, the probability of transferring to any state in the next round can be calculated by conditioning on their current behaviours. By repeating this procedure between any pair of players in all rounds, a transition matrix *M* can be built based on the data of MC in figure 1*c*. The simple Markov model can be described as $S(t) = M^t S(1)$, where *t* is the number of simultaneous rounds (1 simultaneous round equals 7 experimental rounds) and $S(t)$ is the state in round *t*. Using the initial cooperation rate $S(1)$ in

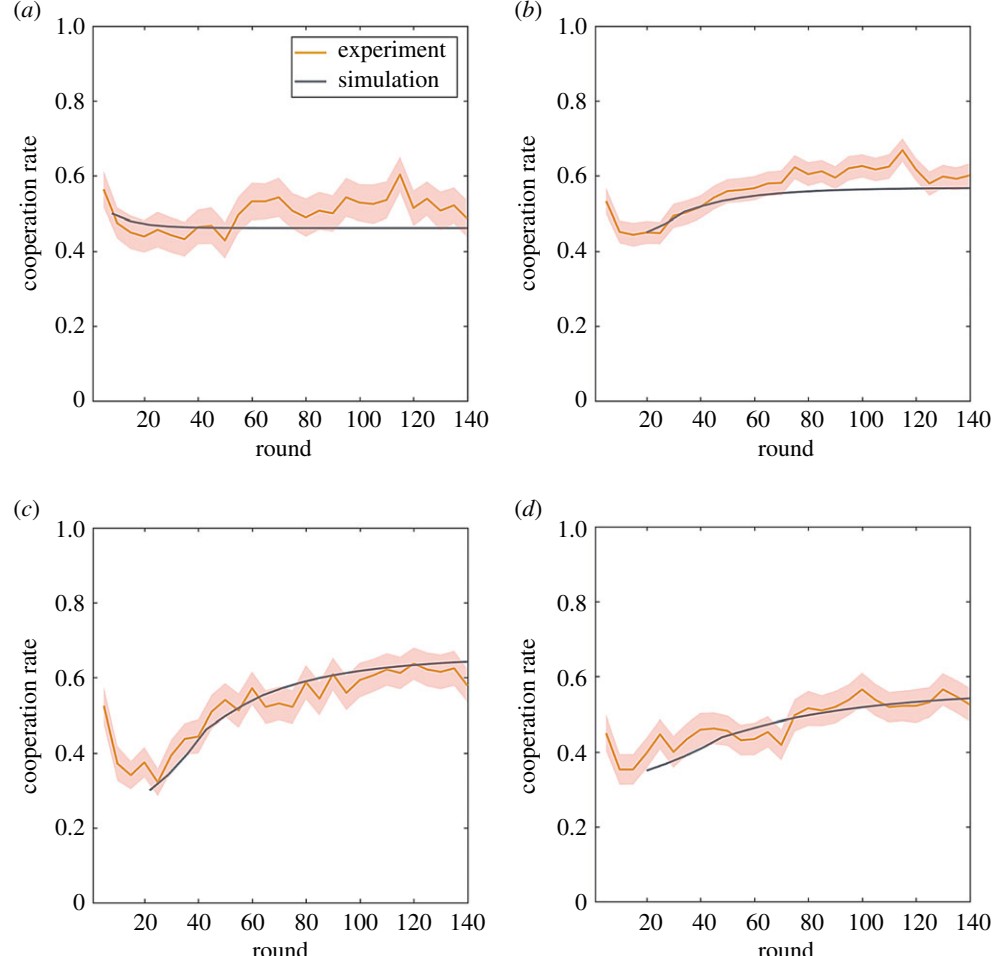

**Figure 5.** Comparison between experimental data and numerical results of a Markov model. Numerical results and experimental data as a function of round number of T1 (*a*), T2 (*b*), T3 (*c*) and T4 (*d*) experiments. The numerical results are obtained from the Markov model, where the transition matrices are obtained based exclusively on the MC in figure 1*c*. The shadow of lines represents mean ± s.e.m.

the experiments, we can calculate the temporal evolution of cooperation in each round and make a comparison with experimental data. Numerical results of simulating the Markov model are in good agreement with experimental results (figure 5), validating our hypothesis that MC is the driving force of cooperation and the evolutionary process can be captured by a Markov process.

# 3. Discussion

We have conducted laboratory experiments to examine the effect of working-memory capacity on cooperative behaviours in repeated PD games. The key to successfully controlling the memory capacity of any subjects lies in preventing them from retaining their own memories and simultaneously showing specified historical actions of others to them. In other words, subjects' memories are reloaded with an adjustable amount of historical information. Our experimental results revealed that memory capacity of 2 leads to the highest cooperation level, which has not been discovered in experiments and predicted by existent models in the literature. Furthermore, there is a sharp increase of cooperation level from the absence of memory to minimal memory capacity, indicating that a minimal memory can ignite the emergence of cooperation. Analyses at the individual level uncovered that subjects employ a MC to make decisions. Evolutionary models based on replicator dynamics and transition matrix are proposed for characterizing evolutionary stable states and temporal behaviours of subjects, respectively. Theoretical predictions and simulations are in good agreement with experimental results.

Long-term memory is involved in relatively steady relationships and interactions with others in relatively small groups. However, in a large population with temporary interactions, short-term (working) memory is unable to be effectively converted to long-term memory for social evaluation. In other words, players may not encounter the same opponents (partners) or game objects any more in a large group. As an extreme case, in a single-shot game, long-term memory does not play a role in making the decision. To be concrete, our experiments only take unstable interactions into account, and short-term memory dominates the behaviours of subjects. Most of the existent models for characterizing cooperative behaviours ignore the short-term memory effect and assume that only one-round historical information is used by agents to make decisions. The ignoring of the memory effect is justified by a mathematical proof that strategies using one-round memory are able to override other strategies using $n$-round memory [48]. However, bounded rational behaviours are manifest in our experiments, differing from the idealized analysis based on evolutionary theory. Furthermore, although a small number of models took the short-term memory effect into account to better characterize decision behaviours [40,43–45], experimental tests on short-term memory are lacking prior to the present work. Interestingly, our experimental findings demonstrated that for the simplest pair interaction, short-term memory plays a pivotal role in determining cooperation behaviour. The effect of short-term memory in our experiments is different from that in previous models with simultaneous interactions in the neighbourhood of each subject [42,46]. Specifically, in our experiments, the setting of a group of randomly encountered subjects aims at controlling their memory capacity, and each subject actually engages in a single-pair interaction with a random partner in every round. Because of the weak indirect reciprocity, our experiment is essentially similar to two-player games. In other words, social ties between each pair of players are independent of others, and every subject actually plays a number of independent two-player games, which differs from the assumption in previous models [42,46]. Even in the simplest scenario without involving networked collective effect as in previous models, the short-term memory still plays a nontrivial role in cooperative behaviours. The inequality of positive and negative information in the short-term memory interacts in a nonlinear manner, influenced by the memory capacity. In the meantime, certain capacity is the prerequisite of reaching a win–win situation in a pair of subjects. The stability of mutual cooperation is as well reflected in the short-term memories and the cost of interrupting a more stable coordination is more unaffordable. The competition between the stronger effect of negative information in memory and the cost of ruining a win–win situation accounts for the non-monotonic cooperative behaviour as a function of memory capacity, and an optimal cooperation level is achieved. We speculate that the integration of social networks with memory capacity will give rise to rich patterns of cooperation, and the optimal memory length may be driven by social learning toward a larger value, as predicted by previous models [42,46]. Additional experimental tests are needed to substantiate the predictions.

Our experimental results provided a possible explanation of the very limited working memory of humans from an evolutionary point of view. Since cooperation among non-relatives was critical for the survival of the ancestors of *Homo sapiens*, any neural functions that facilitate cooperation will be retained by natural selection [22]. According to our results, a very limited short-term memory gives rise to the highest chance to cooperate, implying that the limited working memory of humans may be an evolutionary consequence of the demand for large-scale reciprocity and cooperation. Because an individual with a high working-memory capacity (regardless of its high evolutionary cost) will find it harder to establish a reciprocal relationship and cooperate with others, the individual's fitness will be significantly reduced in spite of some possible advantages of high memory capacity. The negative effect of a high working-memory capacity on cooperation is not the only factor detrimental to fitness. To accommodate more working memories, a bigger brain is needed. But as we know, the brain is a highly energy-consuming organ, such that the brain size is likely to be restricted by evolutionary pressures to optimize fitness. As a result, working-memory capacity is limited as well during the course of evolution, offering an alternative interpretation of limited working memory. Taken together, the genes causing high working-memory capacity of the individual could not spread in the population and limited working memory prevails eventually. It is worth noting that our explanation of limited working memory based on cooperation is not against existent theories in cognitive neuroscience and psychology [34], and our hypothesis is complementary to the understanding of short-term memory limit from an alternative point of view.

Our work raises some questions, and the answers to them may deepen our understanding of memory effect on cooperation and bounded rational behaviours. Note that our experiment is basically a two-player game. If in a group of subjects with multiple interactions, what is the effect of limited working memory on collective cooperation? How do subjects allocate finite memory resources to their group members to make

**Table 1.** The interactive information of focal player ① and his/her partners in rounds 1–7. First row shows that player ① played with player ② in round 1, and both of them chose C. Suppose that the memory capacity of player ① is 4 and player ① will interact with player ② in round 8; the feedback information of player ① is marked in italics in the table.

| Round | Partner ID | Focal player's choice | Partner's choice |
|---|---|---|---|
| 1 | ② | C | C |
| 2 | ③ | C | D |
| 3 | ② | *D* | *C* |
| 4 | ④ | C | D |
| 5 | ② | *D* | *D* |
| 6 | ⑤ | C | D |
| 7 | ④ | D | C |

better decisions? How does limited memory affect other prosocial behaviours, such as punishment, altruism, trust, fairness, etc.? How does limited memory affect coordinate behaviours of human relatives, e.g. apes and other primates? Taken together, our experiments and evolutionary models by incorporating neural basis in terms of limited memory provide insight into cooperation and bounded rational behaviours and suggest that the integration of psychology, behavioural economics, evolutionary games and cognitive neuroscience can advance our knowledge of collective prosocial behaviours.

# 4. Methods

## 4.1. Experimental design

Three hundred and sixty-eight undergraduate students from Beijing Normal University attended the five treatment sessions. Before the experiment started, the rules of the game were explained to all participants (electronic supplementary material, appendix, §3), followed by a quiz to ensure that they fully understood the game. The participants were informed that the experiment would be randomly stopped at 150–200 rounds to avoid the ending effect. The participant was told that he/she was in a group with seven other subjects, his/her partner was randomly changed within group after each round and the interface would show the historical information of his/her and his/her current partner's choices in last $L$ ($L = 0, 1, 2, 3, 4$) rounds. For example, the interactive information of focal player ① and his/her partners in rounds 1–7 is shown in table 1. Suppose that the memory capacity of player ① is $L = 2$ and the player ① will interact with player ② in round 8, the feedback information of player ① will be 'the choices of you and your partner in the last one interaction were D and D' and 'the choices of you and your partner in the second last interaction were D and C'.

## 4.2. The classification of subjects

We classify subjects into four types based on their strategies, namely AC, AD, MC and Other. First, we classify the strategies of subjects in each round into four classes. ① if the strategies used by the focal player in current round and in the memory are all C, the strategy of the focal player in current round is classified in AC, ② if the strategies used by the focal player in current round and in the memory are all D, the strategy of the focal player in current round is classified in AD, ③ if the cooperation rate of focal player's current partner is not less (more) than 0.5, and focal player chose C (D) in current round, the strategy of focal player in current round is classified in MC, ④ for other cases, the strategy of focal player in current round is classified in Other. We note that a strategy can simultaneously belong to more than one category. Second, we classify subjects into four types based on their strategies. The criteria are that if one of the AC, AD or MC is the most frequently used strategy of the subject and the frequency is higher than 2/3, then the subject is marked as the most frequently used strategy; otherwise, the subject will be marked as Other.

Ethics. The experimental protocols adhered to the standards set by the Declaration of Helsinki and were approved by the local Research Ethics Committee at the State Key Laboratory of Cognitive Neuroscience and Learning, Beijing

Normal University, Beijing, China, with reference number CNL_A_0007_001. All participants provided written informed consent to participate after the experimental procedures had been fully explained and acknowledged their right to withdraw at any time during the experiment.

Data accessibility. The data are provided in the electronic supplementary material [51].

Authors' contributions. M.S., Z.Y. and W.W. conceived the study; all authors performed the analysis, discussed the results and wrote the manuscript.

Competing interests. We declare we have no competing interests.

Funding. This work was supported by NNSFC under grant nos. 71631002, 71922004 and 11975049, NSSFC under grant no. 19ZDA062 and NSF DMS under grant nos. 1613035 and 1712714.

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
