## [Peer Review File · Royal Society Open Science]

Review History

RSOS-210653.R0 (Original submission)

Review form: Reviewer 1

Is the manuscript scientifically sound in its present form?

Yes

Are the interpretations and conclusions justified by the results?

Yes

Is the language acceptable?

Yes

Do you have any ethical concerns with this paper?

No

Have you any concerns about statistical analyses in this paper?

No

Recommendation?

Accept as is

Comments to the Author(s)

I thank the Authors for addressing the comments raised in the review for the Proceedings B of the Royal Society. I think the current version explains the contribution of the paper and the experimental design much better than the previous version. I do not have new comments or criticism. One minor comment is that it would be helpful to clarify the parameter values used for Figure 2.

Review form: Reviewer 2**Is the manuscript scientifically sound in its present form?**

Yes

Are the interpretations and conclusions justified by the results?

Yes

Is the language acceptable?

Yes

Do you have any ethical concerns with this paper?

No

Have you any concerns about statistical analyses in this paper?

No

Recommendation?

Accept with minor revision (please list in comments)

Comments to the Author(s)

I only have one suggestion for the empirical analysis and one clarification point:

1. Suggestion for additional analysis: I would suggest to complement the switching rate analysis in Experimental Results with switching rates from D to C. (If I understand correctly, Figure 1d only considers switching from C to D). It is important for the interpretation of the results: the classic explanation for why larger memories might be detrimental for cooperation is that people remember "too much" and do not "forgive". Hence, reestablishing cooperation--that is, switching from D to C--should be decreasing with L. Knowing whether this is the case is key for the understanding of why larger memories lead to lower cooperation rates.

2. Clarification: beginning p.4 - you say that "strategies are shown" but you probably refer to "actions". This should be corrected as there is a difference between the two terms in standard game theory and the word can confuse the understanding of the design.

Decision letter (RSOS-210653.R0)

Dear Dr Zhang

On behalf of the Editors, we are pleased to inform you that your Manuscript RSOS-210653 "Limited memory optimizes cooperation in social dilemma experiments" has been accepted for publication in Royal Society Open Science subject to minor revision in accordance with the referees' reports. Please find the referees' comments along with any feedback from the Editors below my signature.

Please submit your revised manuscript and required files (see below) no later than 7 days from today's (ie 01-Jun-2021) date. Note: the ScholarOne system will 'lock' if submission of the revision is attempted 7 or more days after the deadline. If you do not think you will be able to meet this deadline please contact the editorial office immediately.

on behalf of Dr Feng Fu (Associate Editor) and Pietro Cicuta (Subject Editor)
openscience@royalsociety.org

Associate Editor Comments to Author (Dr Feng Fu):
Both reviewers unanimously recommend acceptance, but there are a few minor changes that are needed. We look forward to receiving your revised manuscript.

Reviewer comments to Author:
Reviewer: 1
Comments to the Author(s)

I thank the Authors for addressing the comments raised in the review for the Proceedings B of the Royal Society. I think the current version explains the contribution of the paper and the

experimental design much better than the previous version. I do not have new comments or criticism. One minor comment is that it would be helpful to clarify the parameter values used for Figure 2.

Reviewer: 2

Comments to the Author(s)

I only have one suggestion for the empirical analysis and one clarification point:

1. Suggestion for additional analysis: I would suggest to complement the switching rate analysis in Experimental Results with switching rates from D to C. (If I understand correctly, Figure 1d only considers switching from C to D). It is important for the interpretation of the results: the classic explanation for why larger memories might be detrimental for cooperation is that people remember "too much" and do not "forgive". Hence, reestablishing cooperation--that is, switching from D to C--should be decreasing with L. Knowing whether this is the case is key for the understanding of why larger memories lead to lower cooperation rates.

2. Clarification: beginning p.4 - you say that "strategies are shown" but you probably refer to "actions". This should be corrected as there is a difference between the two terms in standard game theory and the word can confuse the understanding of the design.

===PREPARING YOUR MANUSCRIPT===

===PREPARING YOUR REVISION IN SCHOLARONE===

Author's Response to Decision Letter for (RSOS-210653.R0)

See Appendix A.

Decision letter (RSOS-210653.R1)

Dear Dr Zhang,

I am pleased to inform you that your manuscript entitled "Limited memory optimizes cooperation in social dilemma experiments" is now accepted for publication in Royal Society Open Science.

on behalf of Dr Feng Fu (Associate Editor) and Pietro Cicuta (Subject Editor)

Appendix A

Dear editor:

Many thanks for your letter. We also thank the two reviewers for the careful reading of the manuscript and the valuable suggestions for improvement. We believe that we have addressed their comments, as we discuss in the following point-by-point response. In addition, all changes in the main text and SI are highlighted by blue color.

With best wishes

Boyu Zhang

Point-to-point response

Reviewer #1

Comment 1: *“One minor comment is that it would be helpful to clarify the parameter values used for Figure 2.”*

Response: We thank the Reviewer for the helpful advice. We added the parameter values “ $\alpha = 0.0969, \sigma = 0.3001$ ” in the caption of Figure 2.

Reviewer #2

Comment 1: *“Suggestion for additional analysis: I would suggest to complement the switching rate analysis in Experimental Results with switching rates from D to C. (If I understand correctly, Figure 1d only considers switching from C to D). It is important for the interpretation of the results: the classic explanation for why larger memories might be detrimental for cooperation is that people remember “too much” and do not “forgive”. Hence, reestablishing cooperation--that is, switching from D to C--should be decreasing with L. Knowing whether this is the case is key for the understanding of why larger memories lead to lower cooperation rates.”*

Response: We thank the reviewer for this pertinent comment. We added the information about the switch rate from D to C of subjects with complete defection history in the third paragraph of Experimental results (lines 143-146):

- In addition, the switch rate from D to C of subjects with complete defection history in T1 is significantly higher than T2-T4, which implies that reestablish cooperation is more difficult for a longer memory length (t-test, see SI Appendix, Table S3 and Table S4).

Furthermore, related statistical results are shown in SI Appendix Tables S3 and Table S4.

Comment 2: *“Clarification: beginning p.4 - you say that “strategies are shown” but you probably refer to “actions”. This should be corrected as there is a difference between the two terms in standard game theory and the word can confuse the understanding of the design.”*

Response: We thank the Reviewer for this valuable comment. We carefully corrected the related terms, where C and D are actions and AC, AD, MC are strategies.